# Effect of Dietary Protein and Tsaa Levels on Performance, Carcass Traits, Meat Composition and Some Blood Components of Egyptian Geese During the Rearing Period

**DOI:** 10.3390/ani10040549

**Published:** 2020-03-25

**Authors:** Elwy A. Ashour, Diaa E. Abou-Kassem, Mohamed E. Abd El-Hack, Mahmoud Alagawany

**Affiliations:** 1Poultry Department, Faculty of Agriculture, Zagazig University, Zagazig 44511, Egypt; elwynutrition@yahoo.com (E.A.A.); dr.mohamed.e.abdalhaq@gmail.com (M.E.A.E.-H.); 2Animal and Poultry Production Department, Faculty of Technology and Development, Zagazig University, Zagazig 44511, Egypt; drdiaaaboukassem_19@yahoo.com

**Keywords:** geese, protein, TSAA, performance, carcass, blood, meat

## Abstract

**Simple Summary:**

Egyptian geese were domesticated more than 4000 years ago; so this bird is among the first domesticated avian species. Therefore, it is necessary to determine the nutritional requirements and establish more precise feeding standards for this breed (Egyptian geese) to ensure its effective production. Dietary level of crude protein (CP) and total sulfur amino acids (TSAA) should closely meet the maintenance and production requirements, especially toward the middle and end of the grow-out period. Thus, this study focused on the effect of varying dietary crude protein and TSAA on growth, carcasses, biochemical blood parameters, and meat quality to determine the nutritional requirement of Egyptian geese. From our results, it can be concluded that the consumption of diets with high levels of protein or methionine and cystine (M + C) can improve the productive performance, carcass and meat quality of Egyptian geese during the rearing period.

**Abstract:**

The present study was performed to investigate the effect of dietary levels of protein, total sulfur amino acids (TSAA), methionine and cystine (M + C) and their interaction on the performance, carcass characteristics, blood components and meat quality of Egyptian geese. A total number of 144 geese at twelve weeks of age were randomly divided into 9 groups (16 birds/each group), each group of birds was sub-divided into 4 replicates, each of 4 birds. There was a significant increase in the bodyweight of geese due to protein and M + C levels (*p* < 0.01). The studied levels of M + C affected significantly on weight gain of growing geese at the early period of 12–18 wk of age. Feed intake was increased with high dietary levels of CP % or M + C (*p* < 0.05). There was a significant (*p* < 0.01) increase in percentages of carcass, liver, dressing, breast and wing with high dietary protein level as compared to a moderate or low level. A high level of dietary protein led to increase in concentrations of total protein and albumin, while total lipids, cholesterol, aspartate aminotransferase (AST) and alanine aminotransferase (ALT) were decreased with increasing level of protein (*p* < 0.01). Fat percentage of breast muscle was significantly (*p* < 0.01) decreased with increasing M + C levels. Protein % of breast muscle was increased with increasing protein levels. Finally, it can be concluded that the consumption of diets with high levels of protein or M + C can improve the bodyweight, feed conversion ratio, carcass and meat composition of Egyptian geese during the rearing period (12–24 wk of age).

## 1. Introduction

The geese are considered to have one of the fastest growth rates of old domesticated birds reared for the production of meat. In 2018, geese and guinea fowl together were kept around the world amounted to 691 million birds, (FAO). Throughout the brooding time, a starter diet of waterfowl in the form of either small crumbles or pellets is recommended. This starter diet normally has a crude protein (CP) ranged between 16% to 18% and a metabolizable energy (ME) of between 10.86 to 12.12 MJ ME/kg [1,2]. Joshi [3] stated that starting at 20-wk-old, goose again begin gaining rapidly throughout the grow-out period. The latest author added that goose is ready for market in 24–30 wk-old. 

The source of dietary protein of high quality with an adequate balance of amino acids is one of the most important factors in feeding Egyptian geese, in particular throughout the rearing phase [4,5]. The supply of nutrients with adequate levels of crude protein and total sulfur amino acids (TSAA) in geese diets through the rearing stage exerts a substantial impact on subsequent reproduction performance. It is important to explore the nutrient requirements of geese and develop more accurate feeding standards for all geese strains to achieve high production. Dietary level of crude protein and amino acid should meet the maintenance requirements and production needs of various poultry kinds, in particular toward the mid and end of the fattening period. Dietary levels of CP had significant impacts on feed intake and feed efficiency of growing Egyptian geese [6]. 

Methionine (Met) amino acid is considered to be the first limiting amino acid in classical feeds used for young birds that play crucial roles in protein structure and anabolism [7]. Met is an essential amino acid for poultry, which used in the structure of the protein and some amino acids. During the metabolism of Met, cystine and homo-cystine are produced. In most cells including the liver, about 50% of cysteine amino acid comes from Met via the trans-sulfuration pathway [8]. The optimal dietary supplementation of methionine could increase growth performance and methionine and cystine utilization in growing goslings [9]. The best strategy to optimize growth, production and overall performance in poultry species as well as mitigating the deleterious effect on the environment is the proper nutrition [10]. Using synthetic amino acids as a feed supplement has greatly a direct effect on growth rates and production as reported by Alagawany et al. [11,12].

In determining the quantities to be used in commercial poultry diets, the criteria used for evaluating the nutritional requirements of Met and TSAA become of concern as mainly measured by performance and feed conversion ratio (FCR). Better productive performance can still be obtained with an adequate level of indispensable amino acids especially Met [13]. Abou-Kassem [14] reported that the highest bodyweight (BW) and bodyweight gain (BWG) was achieved with chicks fed 0.95% TSAA followed by 0.85% in comparison with the control group (0.75%, recommended level, NRC) [15]. Mezes, [16] summarized that amino acid composition of such by-products is far from optimal for poultry whereas, using amino acid supplementation is necessary. Little information has been published describing the amino acid requirements of geese NRC [15].

In brief, most previous studies on the requirements of dietary CP and TSAA have been drawn from the growth performance. The data concerning the effect of dietary CP, TSAA and their interaction on the performance of Egyptian geese during the rearing period are extremely rare. Therefore, the present study was performed to determine the effect of dietary CP and TSAA on performance, carcass characteristics, blood components and meat composition of Egyptian geese during the rearing period (12 to 24 weeks of age).

## 2. Materials and Methods

### 2.1. Birds, Experimental Design and Husbandry 

This work was carried out at Poultry Research Farm belonging to Agriculture College, Zagazig University, Egypt. A total number of 144 Egyptian unsexed geese at twelve weeks of age were randomly divided into nine groups (16 geese/group) each group of geese was sub-divided into four replicates, each of four birds. Each replicate was housed in one pen in-floor system for 12 weeks (experimental period). The area of each pen was 2 m² which provides 0.5 m² per one goose.

A factorial arrangement (3 × 3) was performed including three levels of crude protein (high 16%, moderate 14.5% and low 13%) and three levels of methionine and cystine (0.75, 0.65 and 0.55%) A basal diet was calculated to contain 0.55% total sulfur amino acids (TSAA), and the same diet was supplemented with DL-methionine to provide the level of either 0.65 or 0.75% TSAA for each level of CP, while all diets pelletization was performed using a pellet mill. The feed was conditioned and thermally treated in the fitted conditioners of a pellet mill. Experimental design aimed to study the effect of protein and TSAA levels on the productive performance, carcass traits, meat quality and some blood components of Egyptian geese during the rearing period (from 12 to 24 weeks of age). All experimental diets (Table 1) were formulated according to NRC [15]. All geese were reared under the same managerial conditions, the feed (in pellet form) was given ad-libitum and freshwater provided for the entire goose maintenance period. The environmental conditions was natural for the spring season with temperature degree ranged (18 to 31 °C) while lighting program was 16L: 8D, birds maintained in confinement building of open system for all experimental period.

### 2.2. Data Collection

#### 2.2.1. Performance and Carcass

The geese in all groups were weighed at 12, 18, and 24 wk-old. Daily, feed intake was recorded. Bodyweight gain and feed conversion ratio of geese in all groups were computed. By using standard plastic leg markers individual bodyweight gain was totaled and divided by the number of birds of each replicate to obtain average weight gain in each replicate/group (WG = W_2_ − W_1_). Feed conversion ratio was calculated as the number of grams of feed required to produce one gram of gain during a certain period (feed intake, g/feed intake, g). The mortality rate did not record any bird deaths during the experimental period 12–24 wk of age.

At 24 wk of age, 36 birds (four in each group, two males and two females) were randomly chosen to compute all carcass parameters, then weighed and manually slaughtered by cutting head with a sharp knife to complete bleeding. All carcass parts (eviscerated carcass, heart, liver, breast, wings, gizzard, thighs, neck and back) and abdominal fat were measured according to Blasco and Ouhayoun [17]. Each carcass part weighed, and dressing percentage was calculated by the formula WC/BW 100%, where, WC is the weight of the carcass and BW is bodyweight. The following equation was used: dressed weight = (carcass weight + giblets weight)/live BW.

#### 2.2.2. Blood Parameters 

At 24 wk of age, 36 birds randomly selected (four in each group, two males and two females) to measure only blood parameters. Blood samples were collected at day of slaughter from the sacrificed geese into sterilized tubes. The serum levels of total protein, albumin, lipid profile (total lipids and cholesterol) and glucose as well as activity of alanine transaminase (ALT) and aspartate transaminase (AST) were measured colorimetrically. Albumin, protein, alanine transaminase (ALT) and aspartate transaminase (AST) were assessed using biodiagnostic commercial kits provided from Biodiagnostic Company (29 El-Tahrir St. Dokki, Giza, Egypt) Batch No: ALT (cat#AL1032), AST (cat#AS1062) according to the manufacturers’ guidelines (REF: 264 003, 264 004) and a spectrophotometer (Shimadzu).

#### 2.2.3. Meat Analysis

At 24 wk of rearing period (day of slaughter), sex separated, and after that three males and three females were randomly chosen from eight birds used in previous tests in each experimental group for meat analysis. All selected geese were slaughtered (by cutting head with a sharp knife) after 12 h of feed withdrawal. After 24 h of cooling at +4 °C, post-mortem analyses were carried out. The pH 24 degree of meat samples was determined, briefly, five grams of each meat sample was blended with 45 mL of sterilized water and the pH 24 of the suspension was measured using glass electrode pH meter according to Hussein et al. [18]. The values of pH 24 of breast meat were determined using a digital CyberScan pH-meter 1500 series (Hamilton Company, Reno, Nevada, USA). Meat composition as chemical analysis (moisture, protein, fat and Ash) of breast meat was carried out at Poultry Department, Zagazig University, Egypt according to AOAC [19].

#### 2.2.4. Statistical Analysis

Data (performance, carcass, meat composition and blood parameters) were analyzed on a 3 × 3 factorial arrangement basis using the general liner model procedure in SAS program. The post-hoc Tukey’s test was selected to detect all differences among the treatments (*p* ≤ 0.05), and using the following model: Yijk = *μ*+Ai + Sj+ASij + eijk, where Yijk is an observation, *μ* is the overall mean, Ai is effect of CP level (I = 1–3), Sj is effect of TSAA level (j = 1–3), ASij is the interactions between two variables, and eijk is the experimental random error.

## 3. Results and Discussion

### 3.1. Performance 

Results in Table 2 showed that there was a significant increase in the bodyweight of geese due to protein and M + C levels (*p* < 0.01), where, the high values of bodyweight recorded with the highest levels of protein or M + C levels at 18 and 24 weeks of age. These results agreed with Yang et al. [9] who showed that increasing levels of dietary Met gave a linear increase in bodyweight of Yangzhou geese (*p* < 0.05). Whereas, the interaction between protein and TSAA levels was not significant on geese bodyweight at 12 or 24 wk of age. Dietary protein levels did not affect the weight gain of growing geese during all experimental periods. On the contrast, the body gain was decreased with the decrease of M + C level during 12–18 wk in Table 2. Yang et al. [20] reported that the bodyweight of geese in the 1200 mg/kg methionine group was higher than those in the birds received the control diet.

In other species, such as quail, Abou-Kassem [14] found that bodyweight gain was significantly (*p* < 0.01) for birds fed diet containing methionine level of 100% and 115% over those fed 85% methionine of the NRC [15] recommended level. Bodyweight gain of geese insignificantly affected due to the interaction between protein and M + C levels during all experimental periods (Table 3).

Results in Table 2 showed that feed intake was increased with a high dietary CP % level during experimental periods of 12–18, 19–24 and 12–24 weeks of age (*p* < 0.05). Similarly, high M + C % level significantly (*p* < 0.05) increased feed intake during experimental periods except at period of 19–24 wks of age which was insignificantly affected. Our findings partially agreed with Abou-Kassem [14] who found that feed intake of chicks fed diets containing 100% and 115% of the increased as compared to that contained 85% of the recommended methionine requirements of NRC [15]. On the other hand, Li-Wen et al. [21] reported that feed intake of geese fed diets contained various dietary levels of methionine (0.33, 0.43 and 0.53%) were not significant differences (*p* > 0.05). In parallel, the feed intake of growing geese was not affected by the interaction between CP % and M + C % levels during all periods (Table 3). In Table 2, the feed conversion ratio of geese was significantly (*p* < 0.05) improved with dietary M + C level during 12–18 weeks of age. The main effect of dietary protein or the interaction between protein and TSAA did not affect feed conversion ratio at all ages as shown in Table 2 and Table 3. During 8 to 12 weeks of age, in growing Muscovy ducks, no significant differences in growth performance were observed when CP-diet was reduced from 16 or 15 to 12% based on similar digestible amino acids such as methionine, lysine tryptophan and threonine, Baeza and Leclercq [22]. On the other hand, increasing dietary Met concentration from 0.30 to 0.68% improved bodyweight at 28-d and 35-d in growing Pekin ducks while decreased feed conversion ratio by 7.95% (*p <* 0.05) as a result of nutrient utilization, Zeng et al. [23]. In the present study geese fed on high protein diets had better BW and FCR, which due to relatively high nutrient utilization comes from the nutritive value of diet riches in CP and TSAA especially Met which is an essential and first amino acid in poultry nutrition, and geese body can not synthesis it. Appropriate intake of CP and TSAA were necessary to meet physiological and nutritional requirements for maintenance, body growth of the pre-maturation phase and feathering in geese. Feathers of geese mainly composed of keratin which contains high levels of TSAA percentage and important in the keratin synthesis. The improvement of bodyweight and FCR of geese in this study may due to higher plasma insulin growth factor (IGF-I) concentrations which consists of high dietary CP% as explained by Farhat and Chavez, [24] who observed that Pekin ducks fed on dietary high protein (23%) had higher plasma (IGF-I) concentrations as compared with those fed on dietary levels of 19% or 17% CP, respectively.

### 3.2. Carcass Traits 

A significant (*p* < 0.01) increase in percentages of carcass, liver, dressing, breast and wing was observed with high dietary protein level as compared by a moderate or low level of protein. However, gizzard, giblets and abdominal fat percentages were increased with a low protein diet when compared to the other levels (Table 4). On the other side, heart, thigh, back and neck percentages were not significantly affected by dietary protein levels. Min et al. [25] found that the percentage of carcass, leg and breast meat was highest, but the abdominal fat was lower in geese fed 20% protein than those fed 15% protein. These findings were in agreement with the present study in the percentage of breast and eviscerated carcass but contrary to the percentage of abdominal fat. Our findings partially agreed with Abou-Kassem et al. [6] who observed that dietary level of CP % had significant impacts on carcass traits (*p* < 0.05) but no statistical impacts on percentages of back and dressing whereas, geese fed a high level of CP recorded the highest values of liver, gizzard, heart and giblets compared with those fed other diets which had moderate and low CP% at 12 weeks of age (*p* < 0.05). On the other hand, the percentages of breast meat, carcass, and leg meat were not influenced by different dietary protein levels (*p* > 0.05), while the percentage of abdominal fat increased when dietary protein was 13.54% (*p* < 0.05) [26]. Recently, our findings agreed with Li et al. [27] who clarified that eviscerated carcass and breast yield % increased significantly (*p* < 0.05) with using dietary cassava foliage as a high protein content for Hainan China geese.

In a part from thigh %, dietary levels of M + C had no significant impact on all carcass traits (Table 4). These results in agreement with Neto et al. [28] who summarized that no effect of TSAA levels of (0.659, 0.704, 0.750, 0.796, and 0.841%) was observed on breast, carcass and abdominal fat absolute and relative weight, or for the relative weight of the organs (heart, liver and gizzard) in broilers. On contrarily, EI-Sayiad et al. [29] reported that a positive response of carcass and dressing percentages achieved by increased dietary methionine, where carcass and dressing percentages of quail fed the diet containing 100% and 115% methionine of the NRC recommended requirement was significantly (*p* < 0.01) higher than that birds fed diets containing 85% methionine. The interaction between dietary protein and M + C levels did not significantly affect all carcass traits, except wings % (Table 5). Since the highest value of wings percentage was recorded with the interaction between the high protein and high M + C %. During 8 to 12 weeks of age, no significant differences in carcass quality of growing Muscovy ducks were observed when CP-diet was reduced from 16% or 15% to 12% based on similar digestible amino acids such as methionine, lysine tryptophan and threonine Baeza and Leclercq, [22]. The increasing levels of dietary CP (15%, 17% and 19%) significantly (*p* < 0.01) increased yield of eviscerated carcass and breast meat for growing Pekin ducks, at age of 32 d and 35 d, Zeng et al. [30]. The previous authors added that it is known the CP% and amino acid status in diet influence on the carcass composition of the birds, also in ducks, dietary CP level and amino acids density affected on breast meat yield. The improvement in carcass traits of our findings especially in breast percentage and sequence of dressing percentage agreed with the explanation of Farhat and Chavez, [24] who observed that the ultrasound measurements of breast muscle thickness were 8.42, 7.26 and 6.93 mm for high, medium and low CP% respectively, as well as for the Pectoralis muscle weights as a percentage of carcass weight, 14.38%, 12.19% and 12.02% for high, medium and low CP percentage, respectively.

### 3.3. Blood Constituents

Blood parameters are indicators of body ailments and correlate with the diet quality [31]. Results in Table 6 showed that a high level of dietary protein led to a significant increase in concentrations of total protein and albumin, while total lipids, cholesterol, AST and ALT decreased with increasing level of protein in geese rations (*p* < 0.01). On the other hand, glucose levels did not differ significantly due to dietary protein levels. Our findings partially agreed with Abou-Kassem et al. [7] who concluded that serum protein and its fractions, as well as ALT and AST, significantly differed due to dietary levels of CP. Geese fed rations with 22% CP achieved the highest values of protein and its fractions and the lowest activities of AST and ALT compared with the other groups. On the contrary, Ojediran et al. [31] found that dietary protein levels (22% vs. 16%) did not affect blood biochemical parameters. In broiler flocks, dietary levels of crude protein did not cause any significant alteration in values of cholesterol [32]. 

Irrespective of the dietary protein effect, glucose level was decreased (*p* < 0.01) with increasing of M + C level, while the rest blood biochemical traits did not significantly affect. These results are in disagreement with Yang et al. [19] who found that increasing dietary Met led to a linear increase in serum levels of total protein and its fractions of Yangzhou geese at 70 days of age. The interaction between protein and M + C levels did not affect all blood biochemical traits of geese as shown in Table 7.

### 3.4. Meat Composition 

Regarding pH 24 values of breast meat, there were no significant differences due to protein or M + C levels or their interaction (Table 8). In the available literature, there are few results concerning the conductivity of geese meat and greater differences found between pH measured at 15 min. and 24 h of breast muscles of geese [33]. Marcu et al. [34] concluded that the increase in dietary protein levels significantly increased in pH values of muscle of broiler chickens. In contrast recently, Li et al. [35] reported that using dietary cassava foliage as high protein content for Hainan China geese significantly (*p* < 0.05) decreased the level of pH 24 postmortem. The present study showed that moderate M + C gave the highest value of pH 24, which linked with meat quality as reported by Biesek, et al. [36] who clarified that higher pH of goose meat is often associated with a higher water-holding capacity. On the other hand, the protein percentage of breast muscle significantly (*p* < 0.01) increased with increasing dietary levels of crude protein. Geese chicks from 0 to 4 weeks of age were able to achieve standard growth performance and carcass composition under 20% dietary P level [37]. In the present study, fat percentage of breast muscle significantly (*p* < 0.01) decreased with increasing M + C levels. Similar results were observed in Pekin ducks [23]. In their study, the fat yield and breast skin of Pekin duck were decreased with increasing dietary levels of methionine from 0.30 to 0.68% (*p* < 0.05). According to TSAA levels in present study protein percentage in geese breast insignificantly affected, these findings agreed with Bunchasak et al. [38] who found that no significant differences observed between supplementing TSAA levels of 0.75, 0.94, 1.25, 1.31 and 1.50% in low protein diets of broiler. Our results also were similar to the findings of Conde-Aguilera et al. [39], who found lower content of muscle fat in birds received M + C sufficient ration than received M + C deficient ration. Fat percentage consists on its fatty acids content as Boz et al. [40] who stated that varieties of local Turkish goose reared in extensive system differ in saturated fatty acids, monounsaturated fatty acids and polyunsaturated fatty acids content of breast and thigh meat. In this concern recently, Gumulka and Poltowicz [41] concluded that these differences between Zatorska geese and White Koluda geese may be due to the age, rearing system, and diet differences between the birds. The same previous authors added that the chemical composition of the muscles, except for a 1.35 percentage point higher (*p* < 0.05) dry matter content in the breast muscles of the Zatorska geese than in those of the White Koluda geese which were similar with our findings of Egyptian geese except for higher fat %, it may be due to different in the age of birds in each experiment whereas, fat deposition increased with increasing age of waterfowls. The content of the other nutrients (crude protein, crude fat, and ash) was similar. On the other hand, different levels of dietary protein or TSAA or their interaction did not affect the percentages of moisture and ash of breast muscle as shown in Table 8. In breast muscle, the contents of crude ash, moisture or crude protein were not affected by different methionine levels. However, birds fed the low methionine diets had the highest content of ether extract in among all the treatments (*p* < 0.05) [42].

## 4. Conclusions

The findings of the present study recommended the high levels of CP and M + C for the performance of Egyptian geese during the rearing period to improving the growth performance, carcass traits and meat composition.

## Figures and Tables

**Table 1 animals-10-00549-t001:** Ingredients and chemical composition of the experimental diets (from 12–24 weeks).

CP Level %	16			14.5			13		
M + C Level %	0.75	0.65	0.55	0.75	0.65	0.55	0.75	0.65	0.55
Ingredients (g/k; as–fed basis)									
Corn	665.0	665.0	665.0	687.7	687.7	687.7	710.0	710.0	710.0
Soybean meal (44%)	200.0	201.0	202.0	152.0	153.0	154.0	95.0	96.0	97.0
Wheat bran	96.0	96.0	96.0	120.0	120.0	120.0	152.8	152.8	152.8
Di-calcium P.	16.0	16.0	16.0	16.0	16.0	16.0	16.0	16.0	16.0
Limestone	15.0	15.0	15.0	15.0	15.0	15.0	15.0	15.0	15.0
Premix ^1^	3.0	3.0	3.0	3.0	3.0	3.0	3.0	3.0	3.0
NaCl	3.0	3.0	3.0	3.0	3.0	3.0	3.0	3.0	3.0
DL-Methionine	2.0	1.0	-	2.4	1.4	0.4	2.9	1.9	0.9
L-Lysine	-	-	-	0.9	0.9	0.9	2.3	2.3	2.3
Calculated analysis ^2^									
CP	161.0	160.8	160.5	146.2	146.0	145.9	131.0	130.8	130.3
ME kcal/kg	2810	2806	2803	2815	2812	2810	2813	2810	2807
Ca	10.6	10.6	10.6	10.4	10.4	10.4	10.3	10.3	10.3
Avail. P	4.1	4.1	4.1	4.1	4.1	4.1	4.0	4.0	4.0
Lysine	7.7	7.7	7.7	7.7	7.7	7.7	7.7	7.7	7.7
Met + Cys ³	7.5	6.5	5.5	7.5	6.5	5.5	7.5	6.5	5.5

^1^ Provides per kg of diet: Retinol, (Vit. A) 12,000 I.U; Calciferol, (Vit. D3), 2000 S I; Tocopherol, (Vit. E), 130.0 mg; (Vit. K), 0.67 g; Phytomenadione, 3.605 mg; Thiamin, 3.0 mg; Riboflavin, 8.0 mg; Pyridoxine, 4.950 mg; Cobalamin, 17.0 mg; Niacin, 60.0 mg; D-Biotin, 200.0 mg; Calcium D-pantothenate, 18.333 mg; Folic acid, 2.083 mg; manganese, 100.0 mg; iron, 80.0 mg; zinc, 80.0 mg; copper, 8.0 mg; iodine, 2.0 mg; cobalt, 500.0 mg; and selenium, 150.0 mg.^2^ Calculated according to NRC [15]. ^3^ M + C: Methionine and Cysteine amino acids.

**Table 2 animals-10-00549-t002:** Growth performance parameters of growing geese as affected by dietary treatments.

Traits	CP Levels (%) ^1^			SEM ^3^	*p*-Value ^4^	M + C Levels (%) ^2^			SEM ^3^	*p*-Value ^4^
	High	Moderate	Low			High	Moderate	Low		
Live bodyweight (g)										
12 wk	3102	3090	3098	21.82	0.943	3087	3108	3095	23.68	0.841
18 wk	3681 ^a^	3663 ^a^	3603 ^b^	14.92	0.004	3706 ^a^	3645 ^b^	3597 ^c^	13.78	<0.001
24 wk	4103 ^a^	4067 ^b^	4030 ^c^	18.62	0.040	4102 ^a^	4072 ^b^	4026 ^c^	16.93	0.030
Bodyweight gain (g)/day										
12–18 wk	13.80	13.65	12.02	0.60	0.098	14.73 ^a^	12.79b	11.95 ^c^	0.57	0.013
18–24 wk	10.05	9.62	10.16	0.54	0.787	9.44	10.16	10.23	0.61	0.585
12–24 wk	11.92	11.63	11.09	0.33	0.230	12.09	11.27	11.08	0.29	0.130
Mortality rate (%), 12–24 wk	0.00	0.00	0.00	-	-	0.00	0.00	0.00	-	-
Feed intake (g)										
12–18 wk	150.78 ^a^	148.00 ^b^	143.01 ^c^	2.01	0.041	151.11 ^a^	148.00 ^b^	142.67 ^c^	1.89	0.026
18–24 wk	154.22 ^a^	152.44 ^b^	151.11 ^b^	0.72	0.022	153.30	152.44	151.96	0.68	0.426
12–24 wk	152.50 ^a^	150.22 ^b^	147.06 ^c^	1.20	0.017	152.22 ^a^	149.98 ^b^	147.33 ^c^	0.95	0.033
Feed conversion ratio (kg. feed/kg. gain)										
12–18 wk	10.93	10.84	11.89	0.53	0.184	10.26 ^b^	11.57 ^a^	11.94 ^a^	0.37	0.049
18–24 wk	15.34	15.85	14.87	0.79	0.843	16.24	15.00	14.85	0.69	0.470
12–24 wk	12.79	12.92	13.26	0.41	0.699	12.59	13.31	13.30	0.40	0.453

Means in the same row within each classification bearing different letters are significantly (*p* ≤ 0.05) different.^1^ CP levels: experimental diets, high: 16%, moderate: 14.5%, low: 13%, respectively. M + C ^2^ levels: dietary total sulfur amino acids, high: 0.75%, moderate: 0.65%, low: 0.55%, respectively. ^3^ SEM: standard error mean, ^4^ Overall treatment *p*-value.

**Table 3 animals-10-00549-t003:** Growth performance parameters of growing geese as affected by the interaction among treatments.

	High-CP ^1^			Moderate-CP			Low-CP			SEM ^3^	*p*-Value ^4^
Parameters	High-M + C ^2^	Moderate-M + C	Low-M + C	High-M + C	Moderate-M + C	Low-M + C	High-M + C	Moderate-M + C	Low-M + C		
Live bodyweight (g)											
12wk	3073	3117	3115	3112	3083	3075	3077	3123	3095	8.39	0.858
18 wk	3728	3668	3647	3728	3650	3612	3662	3617	3532	25.85	0.740
24 wk	4157	4123	4030	4092	4060	4050	4061	4032	3998	32.56	0.658
Bodyweight gain (g)/day											
12–18 wk	15.59	13.13	12.66	14.68	13.50	12.78	13.92	11.74	10.40	0.91	0.934
18–24 wk	10.20	10.84	9.13	8.65	9.76	10.44	9.48	9.88	11.11	0.73	0.518
12–24 wk	12.89	11.98	10.89	11.67	11.63	11.60	11.70	10.81	10.75	0.58	0.559
Mortality rate (%), 12–24 wk	0.00	0.00	0.00	0.00	0.00	0.00	0.00	0.00	0.00	-	-
Feed intake (g)/day											
12–18 wk	156.33	151.67	144.33	150.67	149.00	144.33	146.33	143.03	139.00	3.47	0.932
18–24 wk	155.67	154.33	152.70	152.67	150.66	154.00	151.60	152.33	149.30	1.24	0.138
12–24 wk	156.00	153.00	148.50	151.67	149.83	149.17	149.00	147.83	144.33	2.09	0.789
Feed conversion ratio (kg. feed/kg. gain)											
12–18 wk	10.13	11.93	11.75	10.35	11.10	12.01	10.92	12.79	13.98	0.99	0.895
18–24 wk	16.03	14.42	16.83	17.76	15.69	15.03	16.42	15.90	13.78	1.53	0.658
12–24 wk	12.19	12.85	13.68	13.02	12.91	12.97	12.81	13.75	13.49	0.67	0.792

Means in the same row within each classification bearing different letters are significantly (*p* ≤ 0.05) different. ^1^ CP levels: experimental diets, high: 16%, moderate: 14.5%, low: 13%, respectively. M + C ^2^ levels: dietary total sulfur amino acids, high: 0.75%, moderate: 0.65%, low: 0.55%, respectively. ^3^ SEM: standard error mean, ^4^ Overall treatment *p*-value.

**Table 4 animals-10-00549-t004:** Carcass traits and relative weights of growing geese as affected by dietary treatments.

Traits	CP Levels (%) ^1^			SEM ^3^	*p*-Value ^4^	M + C Levels (%) ^2^			SEM ^3^	*p*-Value ^4^
	High	Moderate	Low			High	Moderate	Low		
Eviscerated carcass %	63.99 ^a^	62.40 ^b^	61.05 ^c^	0.29	<0.001	62.80	62.56	62.07	0.31	0.596
Liver %	2.69 ^a^	2.32 ^b^	2.43 ^b^	0.05	0.004	2.55	2.45	2.44	0.04	0.642
Gizzard %	4.54 ^b^	4.73 ^b^	5.28 ^a^	0.09	<0.001	4.95	4.73	4.87	0.11	0.630
Heart %	0.79	0.76	0.76	0.01	0.213	0.79	0.77	0.75	0.01	0.086
Giblets %	8.01 ^b^	7.81 ^b^	8.47 ^a^	0.10	0.016	8.28	7.95	8.06	0.15	0.403
Dressing %	72.00 ^a^	70.21 ^b^	69.63 ^b^	0.27	<0.001	71.20	70.52	70.13	0.29	0.279
Breast %	17.66 ^a^	17.08 ^b^	15.90 ^b^	0.19	<0.001	17.25	16.70	16.69	0.26	0.419
Thighs %	18.86	18.70	18.67	0.13	0.843	18.42 ^b^	18.53 ^b^	19.27 ^a^	0.17	0.012
Wings %	10.21 ^a^	9.14 ^b^	8.92 ^b^	0.18	0.005	9.68	9.55	9.03	0.21	0.321
Back %	7.22	6.50	6.48	0.17	0.134	7.02	6.90	6.29	0.18	0.182
Neck %	5.86	5.99	5.88	0.11	0.301	5.57	5.87	5.60	0.10	0.495
Abdominal fat %	4.84 ^b^	4.99 ^b^	5.31 ^a^	0.07	0.007	4.87	5.08	5.20	0.06	0.112

Means in the same row within each classification bearing different letters are significantly (*p* ≤ 0.05) different. ^1^ CP levels: experimental diets, high: 16%, moderate: 14.5%, low: 13%, respectively. ^2^ M + C levels: dietary methionine and cysteine amino acids level.^3^ SEM: standard error mean, ^4^ Overall treatment *p*-value.

**Table 5 animals-10-00549-t005:** Carcass traits and relative weights of growing geese as affected by the interaction among treatments.

Parameters	High CP Levels (%) ^1^			Moderate CP Levels (%)			Low CP Levels (%)			SEM ^3^	*p*-Value ^4^
	High-M + C ^2^	Moderate-M + C	Low- M + C	High-M + C	Moderate-M + C	Low- M + C	High-M + C	Moderate-M + C	Low- M + C		
Pre-slaughter weight (g)	4157	4123	4030	4092	4060	4050	4061	4032	3998	32.26	0.658
Eviscerated carcass %	65.05	63.54	63.38	62.31	62.57	62.31	61.05	61.58	60.51	0.52	0.283
Liver %	2.73	2.66	2.67	2.40	2.22	2.33	2.51	2.48	2.31	0.13	0.864
Gizzard %	4.34	4.54	4.73	5.07	4.52	4.61	5.44	5.14	5.27	0.19	0.244
Heart %	0.82	0.79	0.76	0.79	0.76	0.74	0.77	0.75	0.76	0.02	0.803
Giblets %	7.88	7.99	8.16	8.26	7.50	7.69	8.71	8.37	8.34	0.26	0.445
Dressing %	72.93	71.53	71.54	70.57	70.07	69.99	70.10	69.95	68.85	0.57	0.737
Breast %	17.51	17.75	17.71	17.65	16.71	16.88	16.59	15.64	15.47	0.39	0.413
Thighs %	18.62	18.07	19.88	18.22	18.62	19.25	18.41	18.91	18.68	0.31	0.054
Wings %	10.36 ^a^	9.96 ^ab^	9.03 ^d^	8.67 ^f^	9.28 ^c^	9.46 ^b^	8.75 ^e^	9.41 ^b^	8.60 ^f^	0.22	0.041
Back %	7.58	7.62	6.47	7.32	6.22	5.95	6.15	6.85	6.44	0.46	0.250
Neck %	4.91	5.34	5.22	5.60	6.01	5.67	6.02	5.55	5.91	0.17	0.092
Abdominal fat %	4.47	4.97	5.08	4.85	5.03	5.10	5.28	5.23	5.42	0.15	0.409

Means in the same row within each classification bearing different letters are significantly (*p* < 0.05) different. ^1^ CP levels: experimental diets, high: 16%, moderate: 14.5%, low: 13%, respectively. ^2^ M + C levels: dietary methionine and cysteine amino acids level. ^3^ SEM: standard error mean, ^4^ Overall treatment *p*-value.

**Table 6 animals-10-00549-t006:** Blood biochemical of growing geese as affected by dietary treatments.

Traits	CP Levels % ^1^			SEM ^3^	*p*-Value ^4^	M + C Levels (%) ^1^			SEM	*p*-Value ^4^
	High	Moderate	Low			High	Moderate	Low		
Total Protein (g/dL)	4.36 ^a^	4.03 ^b^	3.65 ^c^	0.06	<0.001	4.10	4.03	3.91	0.02	0.074
Albumin (g/dL)	1.71 ^a^	1.55 ^b^	1.39 ^c^	0.02	<0.001	1.58	1.54	1.52	0.01	0.250
Total lipids (g/dL)	601.33 ^c^	619.67 ^b^	641.44 ^a^	5.96	0.001	614.67	619.78	628.00	2.91	0.304
Cholesterol (mg/dL)	187.78 ^c^	206.78 ^b^	228.00 ^a^	2.74	<0.001	200.00 ^c^	208.44 ^b^	214.11 ^a^	1.93	0.007
Glucose (mg/dL)	156.67	158.56	158.67	2.91	0.863	156.78	156.00	161.11	1.82	0.427
Aspartate transaminase (U/L)	24.39 ^c^	26.61 ^b^	29.50 ^a^	0.74	<0.001	25.89	27.44	27.17	0.59	0.309
Alanine transaminase (U/L)	29.44 ^b^	29.78 ^b^	33.39 ^a^	0.68	0.001	29.89	31.22	31.50	0.48	0.228

Means in the same row within each classification bearing different letters are significantly (*p* ≤ 0.05) different.^1^ CP levels: experimental diets, high: 16%, moderate: 14.5%, low: 13%, respectively. ^2^ M + C levels: dietary methionine and cysteine amino acids level. ^3^ SEM: standard error mean, ^4^ Overall treatment *p*-value.

**Table 7 animals-10-00549-t007:** Blood biochemical of growing geese as affected by the interaction among treatments.

Parameters	High CP Level % ^1^			Moderate CP Level %			Low CP Level %			SEM ^3^	*p*-Value ^4^
	High-M + C ^2^	Moderate-M + C	Low- M + C	High-M + C	Moderate-M + C	Low- M + C	High-M + C	Moderate-M + C	Low- M + C		
Total Protein (g/dL)	4.49	4.34	4.25	4.11	4.06	3.91	3.70	3.69	3.56	0.08	0.958
Albumin (g/dL)	1.70	1.74	1.68	1.60	1.51	1.52	1.44	1.37	1.36	0.03	0.622
Total lipids (g/dL)	595.67	601.33	607.00	610.00	617.33	631.67	638.33	640.66	645.30	9.41	0.964
Cholesterol (mg/dL)	183.67	189.33	190.33	200.67	204.33	215.33	215.67	231.67	236.60	5.01	0.489
Glucose (mg/dL)	156.33	156.54	157.33	156.30	155.33	164.00	157.67	156.35	162.00	4.29	0.949
Aspartate transaminase (U/L)	23.67	25.17	24.33	25.30	27.30	27.17	28.67	29.83	30.00	1.48	0.988
Alanine transaminase (U/L)	29.33	30.31	28.66	28.00	29.67	31.60	32.33	33.62	34.16	1.57	0.438

Means in the same row within each classification bearing different letters are significantly (*p* ≤ 0.05) different. ^1^ CP levels: experimental diets, high: 16%, moderate: 14.5%, low: 13%, respectively. ^2^ M + C levels: dietary methionine and cysteine amino acids level. ^3^ SEM: standard error mean, ^4^ Overall treatment *p*-value.

**Table 8 animals-10-00549-t008:** Chemical composition of growing geese meat as affected by dietary treatments.

**Traits**	**CP Levels % ^1^**			**SEM ^3^**	***p*-Value ^4^**		**M + C levels (%) ^2^**			**SEM ^3^**	***p*-Value ^4^**
	**High**	**Moderate**	**Low**				**High**	**Moderate**	**Low**		
pH 24	5.84	5.87	5.89	0.03	0.529		5.85	5.91	5.84	0.04	0.403
Moisture %	71.67	72.36	72.31	0.19	0.058		71.68	72.19	72.48	0.22	0.070
Protein %	20.83 ^a^	20.14 ^b^	19.71 ^c^	0.33	<0.001		20.30	20.15	20.23	0.26	0.748
Fat %	5.62	5.61	5.75	0.13	0.719		5.33 ^c^	5.57 ^b^	6.09 ^a^	0.19	0.002
Ash %	1.21	1.20	1.22	0.01	0.815		1.19	1.22	1.20	0.03	0.481
**Parameters**	**High CP Level %**			**Moderate CP Level %**			**Low CP Level %**			**SEM**	***p*-Value ^4^**
	**High-M + C**	**Moderate-M + C**	**Low-M + C**	**High-M + C**	**Moderate-M + C**	**Low-** **M + C**	**High-M + C**	**Moderate-M + C**	**Low-M + C**		
pH 24	5.83	5.86	5.85	5.86	5.89	5.97	5.85	5.94	5.88	0.11	0.633
Moisture %	71.18	71.65	72.17	71.93	72.52	72.74	71.93	72.40	72.62	1.79	0.977
Protein %	20.85	20.78	20.84	20.40	20.05	19.99	19.67	19.63	19.87	0.61	0.741
Fat %	6.09	5.57	5.21	6.11	5.55	5.18	6.08	5.58	5.60	0.39	0.840
Ash %	1.23	1.21	1.20	1.21	1.22	1.19	1.23	1.18	1.21	0.05	0.994

Means in the same row within each classification bearing different letters are significantly (*p* ≤ 0.05) different. ^1^ CP levels: experimental diets, high: 16%, moderate: 14.5%, low: 13%, respectively. ^2^ M + C levels: dietary methionine and cysteine amino acids level. ^3^ SEM: standard error mean, ^4^ Overall treatment *p*-value.

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
