# Peer review of "Effect of Dietary Protein and Tsaa Levels on Performance, Carcass Traits, Meat Composition and Some Blood Components of Egyptian Geese During the Rearing Period"

_animals, 2020, doi:10.3390/ani10040549_

Round 1

Reviewer 1 Report

This paper has been revised and improved obviously according to the review comments, I suggest accept right now.

Author Response

Thank you very much for the reviewer comment and opinion.

Reviewer 2 Report

I don’t think the author did the right statistic methods for analysis, because the table showed in the re submitted manuscript does not change substantively. At least I did not see the interaction effect of CP level and M+C levels.  Maybe a statist could give more suggestions for the statistic analysis.

Question 1:L171-172 The sentence of L171-172 is the same as L176-177. Please check.

Question 2:L174-175 The body gain was decreased with the decreasing of M+C level during 12-18 wk in table 2.

Question 3:L428-430 The reference format.

Question 4:L434-436 Lack the year of publication for reference 29.

Author Response

I don’t think the author did the right statistic methods for analysis, because the table showed in the re submitted manuscript does not change substantively. At least I did not see the interaction effect of CP level and M+C levels.  Maybe a statist could give more suggestions for the statistic analysis.

Thank you very much for your great efforts and give us a chance to improve our paper. We analyzed the data and relay we did not observe any modification than the first one. The data of all parameters were analyzed on a 3 × 3 factorial arrangement basis using the general liner model procedure in SAS program to determine the main effect (Table 2, 4, 6 and 8) of protein, TSAA and their interactions (Table 3, 5, 7, and 8).

Question 1:L171-172 The sentence of L171-172 is the same as L176-177. Please check.

Checked- Deleted.

Question 2:L174-175 The body gain was decreased with the decreasing of M+C level during 12-18 wk in table 2.

Corrected

Question 3:L428-430 The reference format.

Corrected

Question 4:L434-436 Lack the year of publication for reference 29.

Corrected

I highly appreciate the reviewer efforts in my paper.

Reviewer 3 Report

Dear Authors,

your presented manuscript is interesting subject. I could recommend it to publish but it requires major revision. All suggestions are included in attachment in comments form.

Generally:

  • check style of manuscript (editorial mistakes) with instruction for authors (references, titles of subsections)
  • tables should be placed after text between paragraphs.
  • some suggestions in material and methods description
  • in Results and Discussion section Performance and Carcass traits parts could be improved. only comparison with other studies was done. Meat composition was prepared correct (do the same).

To sum up, it's interesting subject and should be publish, but after major revision we could continue.

Kind Regards,

Reviewer

Author Response

Dear Authors,

Your presented manuscript is interesting subject. I could recommend it to publish but it requires major revision. All suggestions are included in attachment in comments form.

Thank you very much for your supportive comments and all your remarks and adjustments have been done

Generally:

  • check style of manuscript (editorial mistakes) with instruction for authors (references, titles of subsections)

Checked and corrected

  • Tables should be placed after text between paragraphs.

Done

  • some suggestions in material and methods description

Done

  • in Results and Discussion section Performance and Carcass traits parts could be improved. only comparison with other studies was done. Meat composition was prepared correct (do the same).

Ok thank you very much for your efforts and adjustments. We added some paragraphs to improve the discussion of Performance and Carcass traits. Thanks again for your efforts and giving us a chance to improve our paper.

To sum up, it's interesting subject and should be publish, but after major revision we could continue.

Also, all reviewer comments in the Pdf file have been adressed in the revised version all responses also added in the attached pdf.

Thank you very much for your supportive comment

Kind Regards,

Round 2

Reviewer 3 Report

Dear authors, 

Thank you, I have no more questions

Kind Regards 

This manuscript is a resubmission of an earlier submission. The following is a list of the peer review reports and author responses from that submission.

Round 1

Reviewer 1 Report

The paper is interesting, well written, although it has quite some defects. The paper requires correction before publishing.

At first use, give the full name plus its abbreviation (TSAA,M+C). 

P value should be in italics.

Introduction

Please present production and nutrient requirement (especially protein requirement )of goose in Egypt. Explain why not use NRC recommendations for geese.

Provide the data on goose meat production in the world, including goose meat production in Egypt according to FAOSTAT website data (2018) – http://www.fao.org/faostat/en/#data/QL.

The mean of this sentence “The geese have the fastest growth rate of birds reared for production of meat” is not correct. How about growth rate of geese compare with turkey, ostrich, and emu.

Materials and Methods

M+C respective adding proportion?

The number of geese used for research is sufficient, but the number of gander in the subgroup is small (n = 4).

Mortality should be list in Performance and carcass.

Results and discussion

Comparison of the performance between Egyptian geese and geese from other countries(China,France,Germany etc) (e.g. Li et al. 2019, 2020 see Poultry Science publication concerning cassava fed to geese in China).

Mortality should be list in Table 2.

Is the amino acid composition of muscle analyzed? If so, please add it to table 8, CP and M+C May affect the amino acid content of muscle.

Conclusion

The conclusion part is more detailed, including the specific data recommended.

Reviewer 2 Report

Levels of protein and sulfur amino acids are important factors in dietary formula of geese breeding. Authors designed an 3×3 experiment to investigate the effect of dietary protein and TSAA levels on performance, carcass traits, meat composition and blood components of Egyptian geese. However, I have the following comments and questions on the manuscript which the authors should consider. 

The authors used 12 weeks unsexed geese for the experiment, but when slaughter for blood sample collection, the author chose 4 geese in each group (line 129), while when the meat analysis sample collection, 3 males and 3 females were chosen for each group (line 135-136). How many geese were chosen for each group totally in this experiment? It is inconsistent in the methods section. Actually, it should use male and female half or entire males for this experiment.

As to the 3×3 experiment, the interaction effect and main effect of two experimental factors should be considered in the statistic analysis. Authors should use PROC GLM procedure in SAS.

Discussion: The results and discussion section should be well-discussed for results and interpretation with previous related studies. But this manuscript simply listed the experimental results of others.

Table 1: The column of nutrient levels were all calculated. The author should measure feeds protein and M+C levels at least.

Table 2: FCR cannot be calculated by Feed intake and Body weight gain. Please list the calculation formula and check the data.

L110-111: Is the formula not include vitamin A / D2 / E / K1?

L129-133: Please add information about the instrument and manufacturer.

L157-158: The description does not match the table 2.

L164: P<0.05 or P≤0.05? The same as below.

L190: P<0.05 or P>0.05?

L196-202: The contents are similar to L232-238. Please explain the results of Zeng's research.

L253-254: The description does not match the table.

Table 8: pH instead of Ph

L325-326: Reference [6] does not match the content of L64-66, Please check.

L350-352 Reference format is incorrect. 

L354: Delete”

L360: Delete“

Reviewer 3 Report

Dear Authors,

your presented manuscript is interesting subject, especially there is geese rearing, but I have to many suggestions to be written here.

Overview:

The paper is not prepared for Animals journal at all. Everything needs to be improve.
Methods are descirbed too short. more precisely. I can't know which aparatures were used.
Discussion is also too short. There is more scientific papers in goose and protein content subjects, for example: Alternatives Protein in Animal Nutrition in Animals journal, or the other works presented by many researchers, in geese or ducks subject. So it is connected with references. Only 32...
At this moment I can't recommend your manuscript to publish. After major revision you should try submit again. It could be accepted after major improvement, because goose rearing is unpopular and should be promoted. 

Check format style precisely! Empty pages, letters etc...

Kind Regards,

Reviewer
